# Autothermal Reforming of Methane: A Thermodynamic Study on the Use of Air and Pure Oxygen as Oxidizing Agents in Isothermal and Adiabatic Systems

Matheus Henrique Silva Cavalcante [1], Ícaro Augusto Maccari Zelioli [2], Emílio Émerson Xavier Guimarães Filho [2], Julles Mitoura dos Santos Júnior [2,*], Annamaria Dória Souza Vidotti [1], Antonio Carlos Daltro de Freitas [1] and Reginaldo Guirardello [2]

[1] Engineering Department, Exact Sciences and Technology Center, Federal University of Maranhão (UFMA), Av. dos Portugueses, 1966, Bacanga, São Luís 65080-805, MA, Brazil; matheus.cavalcante@ufma.discente.br (M.H.S.C.)

[2] School of Chemical Engineering, University of Campinas (UNICAMP), Av. Albert Einstein 500, Campinas 13083-852, SP, Brazil

* Correspondence: jullesmitoura7@gmail.com; Tel.: +55-(98)-991060212

**Abstract:** In this paper, we analyze the autothermal reforming (ATR) of methane through Gibbs energy minimization and entropy maximization methods to analyze isothermic and adiabatic systems, respectively. The software GAMS® 23.9 and the CONOPT3 solver were used to conduct the simulations and thermodynamic analyses in order to determine the equilibrium compositions and equilibrium temperatures of this system. Simulations were performed covering different pressures in the range of 1 to 10 atm, temperatures between 873 and 1073 K, steam/methane ratio was varied in the range of 1.0/1.0 and 2.0/1.0 and oxygen/methane ratios in the feed stream, in the range of 0.5/1.0 to 2.0/1.0. The effect of using pure oxygen or air as oxidizer agent to perform the reaction was also studied. The simulations were carried out in order to maintain the same molar proportions of oxygen as in the simulated cases considering pure oxygen in the reactor feed. The results showed that the formation of hydrogen and synthesis gas increased with temperature, average composition of 71.9% and 56.0% using air and $O_2$, respectively. These results are observed at low molar oxygen ratios ($O_2/CH_4 = 0.5$) in the feed. Higher pressures reduced the production of hydrogen and synthesis gas produced during ATR of methane. In general, reductions on the order of 19.7% using $O_2$ and 14.0% using air were observed. It was also verified that the process has autothermicity in all conditions tested and the use of air in relation to pure oxygen favored the compounds of interest, mainly in conditions of higher pressure (10 atm). The mean reductions with increasing temperature in the percentage increase of $H_2$ and syngas using air under 1.5 and 10 atm, at the different $O_2/CH_4$ ratios, were 5.3%, 13.8% and 16.5%, respectively. In the same order, these values with the increase of oxygen were 3.6%, 6.4% and 9.1%. The better conditions for the reaction include high temperatures, low pressures and low $O_2/CH_4$ ratios, a region in which there is no swelling in terms of the oxygen source used. In addition, with the introduction of air, the final temperature of the system was reduced by 5%, which can help to reduce the negative impacts of high temperatures in reactors during ATR reactions.

**Keywords:** autothermal reforming of methane; Gibbs energy minimization; entropy maximization

## 1. Introduction

As the search for clean and renewable energy sources becomes more and more imperative, there is a substantial growth in the demand to produce hydrogen, synthesis gas and other fuels that are produced from renewable sources. This is becoming a preponderant topic in contemporary discussions and research, in order to diversifying the global energy matrix. In 2021, about 47% of the hydrogen produced globally was derived from natural gas, while the rest was produced from coal (27%), oil (22%) and electrolysis (4%) [1,2].

In this scenario, it is important to highlight the thermochemical transformation processes as a viable technology for generation of hydrogen from natural gas. Thermochemical transformation is used in the conversion of a certain fuel consisting of hydrocarbons to generate a gas with a composition that has a high content of hydrogen (for pure hydrogen production) or a combination of hydrogen and carbon monoxide (for syngas production), used for various petrochemical processes [2–4]. In the United States, more than 95% of hydrogen is generated by steam reforming of methane (SMR), producing 10 million tons of this product annually [5], with the process being responsible for more than 50% of global hydrogen production [6]. This technology uses the reaction of methane with steam at high temperatures to form hydrogen and carbon monoxide, the latter of which is able to subsequently react with steam to produce more $H_2$.

Other thermochemical routes can be used for this purpose, such as partial oxidation (PO) and autothermal reform (ATR). Among the cited alternatives, the autothermal reform of methane, which is of great importance for many applications related to chemical conversion for energy generation, is mainly used in large conversion units [7]. This reaction variant joins steam reforming and oxidative reforming reactions, emerging as a very promising alternative. This is especially due to the more interesting thermal behavior, compared with the steam reforming reaction, which is essentially endothermic, considering that it is also a clean and efficient process [8].

The costs for each technology used for hydrogen production vary in relation to efficiency, scale of production and implementation of carbon capture and storage (CCS) technologies [9], as shown in Table 1. $CO_2$ capture is incorporated, ATR has benefits since the most efficient separation process for this case manages to recover carbon dioxide at 3 atm, reducing the costs associated with compression to achieve pipeline transport pressures [7].

**Table 1.** Cost comparison between the SMR and ATR processes [10].

| Technology | Capital Cost (M$) | $H_2$ Production Cost ($/kg) |
|---|---|---|
| SMR with CCS | 226.4 | 2.27 |
| ATR with CCS | 180.7 | 2.08 |
| $CH_4$ ATR with CCS | 183.8 | 1.48 |

Some studies have already discussed the process of autothermal reform, such as the work by Rau et al. [11] who studied the efficiency of a pilot plant for hydrogen production through autothermal reforming of biogas using air under different temperatures and ratios of oxygen to carbon. Results showed that higher efficiencies were found at $O_2/CH_4$ ratios ranging between 0.8 and 0.9 over a range of 773–973 K.

Yan et al. [12] investigated the characteristics of autothermal methane reforming using the thermodynamic equilibrium constant method, demonstrating the best operating conditions, showing that the reaction occurs more easily at 1000 K and air/methane and steam/methane ratios of one and two, respectively.

Sayar and Eskin [3] evaluated the effects of inlet conditions on the performance of an autothermal natural gas reforming reactor using a monolithic catalyst and air as an oxygen source, discussing the kinetics and thermodynamics of the reaction. Although much studied, this reaction is still not completely understood, especially about its thermal behavior throughout the process.

Recent studies have shown a greater focus on characterizing the thermal behavior of ATR processes. In Cherif et al. [13] an ATR reactor was designed and optimized aiming high $H_2$ yields. The authors report important and significant results in improving thermal and $H_2$ selectivity of the optimized reactor. The configuration improved the performance compared to the traditional model: the highest average temperature was reduced by 24.8%, while the methane conversion improved by 27.2%.

In the work they developed, Hu et al. [14] reported a computational fluid dynamics study to simulate a base case of the microchannel reactor that couples hydrogen catalytic

combustion with methanol steam reforming. The combined proposal for the process demonstrated operational and technical feasibility by reducing the formation of hot spots in the reactor and improving the use of the catalyst in the process.

Although many important studies have been conducted on the methane ATR process, some gaps still need to be filled to fully understand the thermodynamic characteristics and the thermal behavior of this reaction. An effective way to promote the understanding of reaction pathways during methane ATR reaction appears through the application of thermodynamic analysis methodologies, such as Gibbs energy minimization and entropy maximization methods [15].

In previous research by our group, some studies were carried out to understand the behaviors associated with methane valuation processes using thermochemical routes. In Mitoura et al. [1], the process of thermal cracking of methane was studied, aiming to produce pure hydrogen. In Freitas and Guirardello [15], the methane ATR process was studied (the same process studied in the present work); however, in Freitas and Guirardello [15], the thermodynamic data were constructed considering the ideality of the gas phase, a factor that brings a certain limitation to the analyzes conducted and consequently affects the range of application of the results obtained.

Likewise, in this work a thermodynamic analysis of autothermal reforming (ATR) of methane was conducted. The Gibbs energy minimization method was used to calculate the reaction equilibrium compositions under constant pressure and temperature conditions, using both air and pure oxygen as the oxygen source. To calculate the equilibrium temperature of the system under constant pressure and enthalpy condition, the entropy maximization method was used. For both cases, the non-idealities were determined using Virial equation of state to determine the fugacity coefficient of gas phase. Throughout the text, the influence of reaction parameters will be verified, such as pressure, temperature and feed composition, and we will analyze the thermal effect during the use of air as an oxidizer during ATR of methane.

The differentiation proposed by the thermodynamic model developed in this work is use of the virial equation of state (EoS) as an estimator of non-idealities for the gas phase throughout the methane ATR process. The use of this EoS has proven to be robust and reliable in other studies for different systems in the literature [1,16,17]. However, none of these works consider the application of these methods in the study of the ATR of methane process.

## 2. Methodology

### 2.1. Gibbs Energy Minimization: Isothermal Reactors

The equilibrium composition can be determined for a system with multiple components and phases, at constant pressure and temperature, by minimizing the Gibbs energy of the system considering the number of moles of each component in each phase. Equation (1) represents this for a system composed of a gas, a liquid and a solid phase [15].

$$min\ G = \sum_{i=1}^{NC} n_i^g \mu_i^g + \sum_{i=1}^{NC} n_i^l \mu_i^l + \sum_{i=1}^{NC} n_i^s \mu_i^s \tag{1}$$

The restrictions for the model are found in the non-negativity of the number of moles (Equation (2)) of each component in each phase and the balance of moles obtained by the atomic balance for reactive systems (Equation (3)).

$$n_i^g, n_i^l, n_i^s \geq 0 \tag{2}$$

$$\sum_{i=1}^{NC} a_{mi}(n_i^g + n_i^l + n_i^S) = \sum_{i=1}^{NC} a_{mi} n_i^0,\ m = 1, \ldots, NE \tag{3}$$

The indices *g*, *l* and *s* represent the gaseous, liquid and solid phase, respectively, while $n_i$ and $a_{mi}$ are the number of moles for each component and atoms of each element in a

molecule, respectively. *NC* and *NE* are the number of components and types of atoms in the system, in that order. The Gibbs energy was calculated considering that the components were in only one gaseous phase and there was no coke formation. These considerations were used in previous research with good results [16]. Equation (4) represents the Gibbs energy with these considerations.

$$G = \sum_{i=1}^{NC} n_i^g \left( \mu_i^g + RT \left( \ln P + \ln y_i + \ln \phi_i \right) \right) \tag{4}$$

Non-ideality was represented by the fugacity coefficient, calculated by truncated virial state equations in the second coefficient. The equation for the second virial coefficient was based on Pitzer's correlation [18] modified by Tsonopoulos [19]. The equation is shown as Equation (5).

$$\ln \hat{\phi}_i = \left[ 2 \sum_{j}^{m} y_j B_{ij} - B \right] \frac{P}{RT} \tag{5}$$

Since $\mu_i^g$ and $y_i$ are the chemical potential and mole fraction of the component, $R$ is the gas constant, $T$ is the temperature of the system, $P$ the pressure, $\phi_i$ and $\hat{\phi}_i$ the fugacity coefficients of the pure component and in the mixture, $m$ is the atom in a molecule; $B$ is the second coefficient of the virial and $B_{ij}$ is this cross coefficient.

The use of the virial equation of state to incorporate non-idealities in the thermodynamic model studied in this research improves predictions of equilibrium compositions for thermochemical process and represents a crucial advancement in the field of research on reaction pathways for fuel production in comparison with previous results presented in the literature.

By accounting for deviations from ideal gas behavior, it is possible to predict and analyze equilibrium compositions with greater accuracy and depth, even under moderate pressures and in complex reactions systems. Furthermore, the utilization of entropy maximization methodology provides a comprehensive approach to the system, highlighting the impact of initial process conditions on the final temperature of the output stream. This is useful for comparing the influence of the oxidizer in equilibrium compositions and equilibrium temperatures during ATR of methane process.

*2.2. Entropy Maximization: Adiabatic Reactors*

Thermodynamic equilibrium can also be studied by maximizing the entropy of the system at constant pressure and enthalpy as presented in Equation (6).

$$max \ S = \sum_{i=1}^{NC} n_i^g S_i^g + \sum_{i=1}^{NC} n_i^l S_i^l + \sum_{i=1}^{NC} n_i^S S_i^S, \qquad n_i^g, n_i^g, n_i^g \geq 0 \tag{6}$$

The restrictions mentioned above (Equations (2) and (3)) are repeated for the entropy maximization model, with the addition of conservation of enthalpy and non-negativity of absolute temperature (Equations (7) and (8)).

$$\sum_{i=1}^{NC} (n_i^g H_i^g + n_i^l H_i^l + n_i^S H_i^S) = \sum_{i=1}^{NC} n_i^0 H_i^0 = H^0 \tag{7}$$

$$T \geq 0 \tag{8}$$

In this case, $H_i$ is the partial molar enthalpy of the component, $H_i^0$ is the partial molar enthalpy of input of the component, and $H$ is the total enthalpy of the system.

In this model, the non-ideality of the vapor phase was also considered through the use of the virial equation, by calculating the fugacity coefficient through Equation (5).

The deviations of the models in relation to the literature data were calculated using the relationship below, Equation (9).

$$Deviation\ (\%) = 100 \times \frac{\left| x_i^{lit.} - x_i^{calc.} \right|}{x_i^{lit.}} \tag{9}$$

Methane conversion and hydrogen selectivity were obtained by Equations (10) and (11).

$$X_{CH_4} = \frac{n_{CH_4,in} - n_{CH_4,out}}{n_{CH_4,in}} \tag{10}$$

$$S_{H_2} = \frac{n_{H_2,out}}{n_{CO,out} + n_{CO_2,out}} \tag{11}$$

The thermodynamic analysis of methane autothermal reforming covered pressures of 1, 5 and 10 atm, initial temperatures of 873, 973 and 1073 K, oxygen/methane molar ratios (OCR) between 0.5/1 and 2/1, and ratios of air/methane molars (ACR) from 2.5/1 to 10/1 (with 80% $N_2$ and 20% $O_2$). The vapor/methane ratio was analyzed in the range of 1.0/1.0 and 2.0/1.0 during all conditions tested in this paper. The operational conditions were determined based on previous literature [3,20], and represent the main operational testing range for the methane ATR process.

The models used perform simultaneous chemical and phase equilibrium calculations and are a non-linear programming problem. To carry out the simulations, the GAMS (General Algebraic Modeling Systems)® 23.9.5 software was used in combination with CONOPT3 solver. This solver is based on the concept of generalized reduced gradient, a reliable algorithm for solving non-linear programming problems, such as the Gibbs energy minimization and entropy maximization methods proposed by the present work. This combination of software and solver was previously used by our research group with excellent results [1,15,17]. A total of 12 compounds were considered during simulations including $H_2$, $CH_4$, $H_2O$, $CO$, $CO_2$, $O_2$, $N_2$, $NH_3$, $NO$, $NO_2$, $CH_3OH$ and $C_2H_6$. All thermodynamic properties of considered compounds were obtained in Polling et al. [21].

## 3. Results and Discussion

The generation of hydrogen and syngas was examined at various pressures, temperatures and inlet compositions in the context of autothermal methane reforming (AMR). In the subsequent section, the proposed Gibbs energy minimization model was validated through comparison with experimental and simulated data from the literature, referring to the investigated process.

### 3.1. Gibbs Energy Minimization Methodology Validation

The results of the Gibbs energy minimization model were compared with the model data presented by Carapellucci and Giordano [22], and the experimental data presented by Lutz et al. [23] who studied three methane reforming processes: steam reforming, dry reforming and autothermal methane reforming, comparing literature data with our equilibrium model based on the minimization of Gibbs energy. Below are graphs (Figure 1) comparing the model used in the simulations performed by this paper with the experimental data from Lutz et al. [23] and those simulated by the model proposed by Carapellucci and Giordano [22] for the methane steam reforming process.

The calculated deviations were small and the model proved to be well suited for predicting the gas compositions at the reactor outlet at the different temperatures employed during experimental investigation. The results presented in Figure 1 were obtained under a pressure of 10 atm and at constant $H_2O/CH_4$ ratio of two.

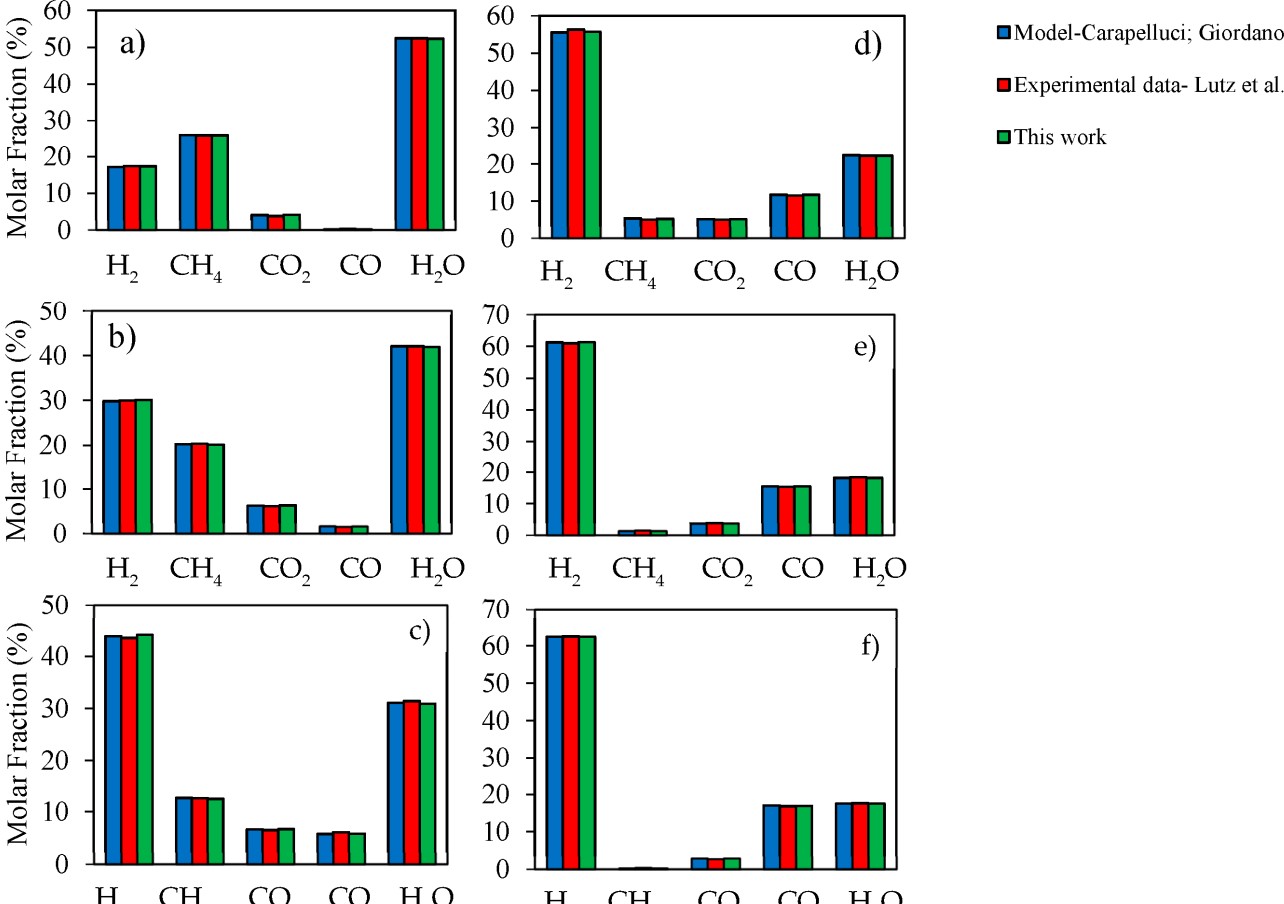

**Figure 1.** Results obtained from the Gibbs energy minimization model used in this work and the data simulated by Carapellucci and Giordano [22] and experimental data from Lutz et al. [23]: (**a**): 773 K; (**b**): 873 K; (**c**): 973 K; (**d**): 1073 K; (**e**): 1173 K; (**f**): 1273 K.

The differences were, in general, less than 5%, except for some lower molar fractions. The satisfactory agreement between the equilibrium data and the experimental data confirms the ability of the proposed model to provide consistent results. The results showed that hydrogen production is favored with increasing temperature, which was already expected due to the fact that the methane steam reforming process is endothermic, which is favored with this change.

These results represent an advance in relation to the ideal model proposed by Freitas and Guirardello [15] during ATR of methane. The authors, who considered the ideality of the phase, report predictions that were more distant from the experimental data than that observed by this paper. Deviations on the order of 6.5% were reported considering ideal behavior for gas phase.

Methane conversion and hydrogen selectivity for autothermal methane reforming at 1 atm and a fixed molar ratio $O_2/CH_4$ of 0.1/1.0 at different temperatures were compared with experimental data from Ayabe et al. [20], as shown in Figure 2. As with the results for steam reforming, the deviations were small, generally below 1.0%. The increase in methane conversion with temperature occurs due to the increase in production, especially of hydrogen, in addition to fractions of by-products, such as carbon monoxide and carbon dioxide, and ammonia is also present in small amounts due to the presence of nitrogen in the feed stream.

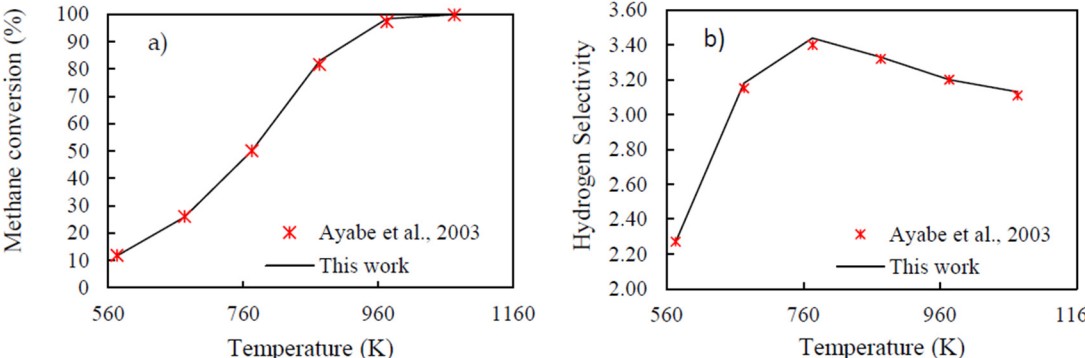

**Figure 2.** Comparison of the results of the Gibbs thermodynamic energy minimization model and the experimental data of Ayabe et al. [20]: (**a**) methane conversion; (**b**) hydrogen selectivity.

At lower temperatures, the equilibrium composition will contain more methane and water that did not react for the formation of hydrogen, considering that under these conditions the CO methanation reaction may occur more, evidently reducing the amount of hydrogen for the formation of hydrogen, methane and water [24]. Ayabe et al. [20] also report that the methane conversion rate as a function of temperature begins to decrease at values above 773 K. Thus, the selectivity of hydrogen does not show a continuous increase with increasing temperature, following the behavior of the conversion at equilibrium. Furthermore, according to Rau et al. [11], the main reactions for autothermal methane reforming are:

$$CH_4 + \frac{1}{2}O_2 \leftrightarrow CO + 2H_2 \ \Delta H_{298K} = -35.6 \frac{kJ}{mol} \tag{12}$$

$$CH_4 + H_2O \leftrightarrow CO + 3H_2 \ \Delta H_{298K} = +206.2 \frac{kJ}{mol} \tag{13}$$

$$CO + H_2O \leftrightarrow CO_2 + H_2 \ \Delta H_{298K} = -41.2 \frac{kJ}{mol} \tag{14}$$

Observing the reactions above, we note that the steam reforming reaction is the one that individually has the greatest amount of moles of hydrogen produced, and also contains the greatest heat variation among the others. This indicates that the effect of temperature on the endothermic reaction of steam reforming counterbalances the opposite behavior of increasing this parameter on exothermic reactions, leading to a net increase in hydrogen formation. In addition, these opposite effects of temperature in the reactions can help to better control the thermal behavior of the process.

### 3.2. Thermodynamic Analysis for Isothermal System Reaction

The simulation of autothermal methane reforming was carried out under two conditions: in the first, only pure oxygen was used as the oxidizing reagent, and in the second, atmospheric air (with 80% nitrogen and 20% oxygen) was used. The pressure range from 1 to 10 atm, the initial temperature from 873 to 1073 K, the $O_2/CH_4$ ratio from 0.5 to 2.0 and the $H_2O/CH_4$ ratio was fixed at 1.0. Coke formation was not allowed in the simulation and 1 mol of $H_2O$ and $CH_4$ was used in the feed. Figure 3 shows how hydrogen production is affected by pressure, process temperature and oxidizer.

Analyzing the behavior of the graphs shown in Figure 3, it is possible to notice the greater production of hydrogen for the condition of the air entering the reaction. This is due to the reduction of the partial pressure of the reactants when nitrogen is added, favoring the balance of the steam reforming process for the formation of hydrogen, according to Le Chatelier's principle, obtaining a higher conversion of $CH_4$ in the reaction. Similar results are reported in previous works of literature [25]. This behavior was consistently repeated for the other analyzed $O_2/CH_4$ ranges.

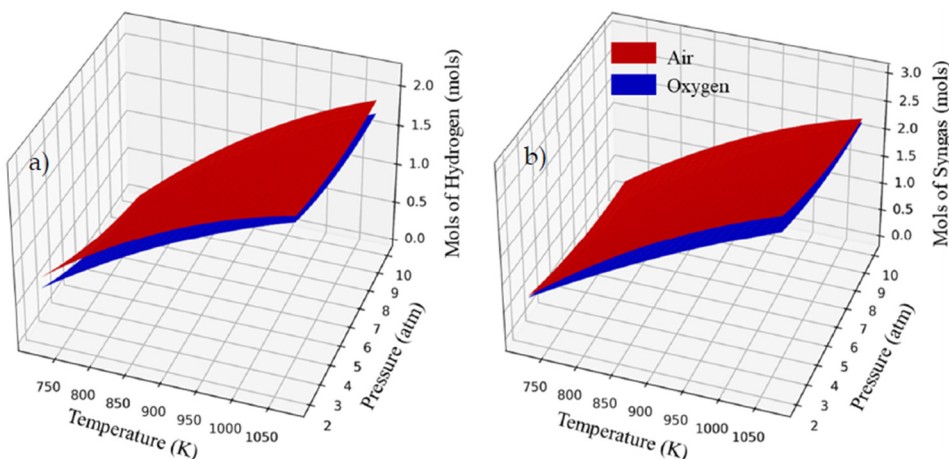

**Figure 3.** Moles of hydrogen (**a**) and synthesis gas (**b**) under different temperatures and pressures for $O_2/CH_4 = 0.5$ and $H_2O/CH_4 = 1.0$.

The increase in the amount of oxygen in the process for $O_2/CH_4$ ratios greater than 0.5 showed a decrease in hydrogen production, which influenced the formation of synthesis gas, which was already expected. This behavior is practically independent of the used oxidizer, noting that the decline for the two cases studied is approximately the same. In view of this, Freitas and Guirardello [15] indicate that for ratios below 0.5 the addition of $O_2$ does not have much impact on $H_2$ production. While for larger ratios, the amount of this product starts to decrease, corroborating the results obtained and observed in Figure 4.

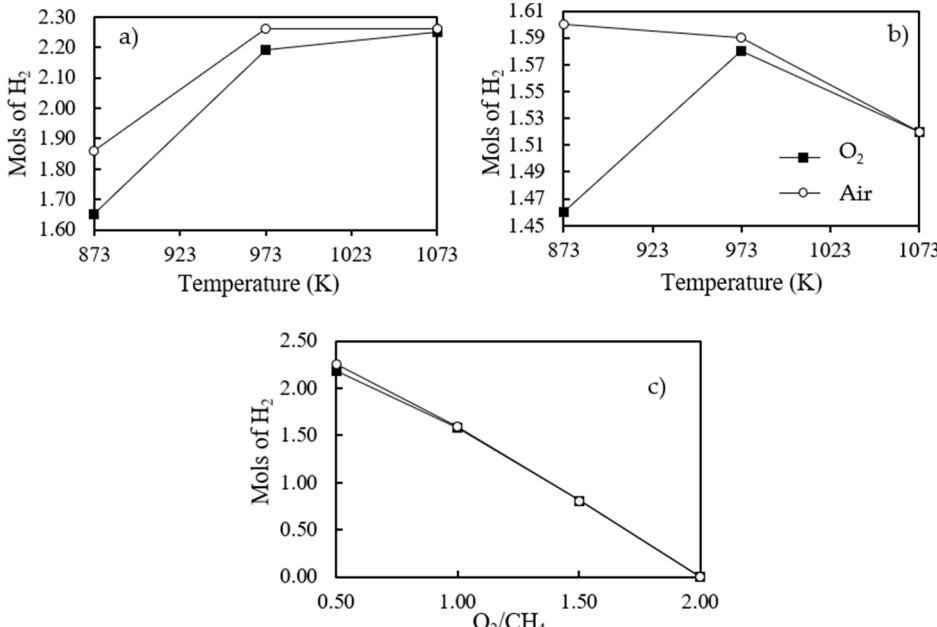

**Figure 4.** Hydrogen production at different temperatures and oxygen ratios, at 1 atm: (**a**) OCR = 0.5; (**b**) OCR = 1; (**c**) 973 K.

The negative effect of temperature that begins to be evident with the increase of oxygen in the system can be changed with the increase of pressure; however, this would decrease the number of moles formed of hydrogen and synthesis gas. On average, the production of these compounds was reduced by 19.7% using $O_2$ and 14.0% using air, by intensifying the pressure in the different investigated conditions. The increase in hydrogen production with the use of air as an oxidizing agent in relation to the use of pure oxygen became more noticeable as the pressure was increased, demonstrating a behavior of the system using air

less susceptible to a drop in hydrogen production due to the pressure increase. The trend is shown in Figure 5.

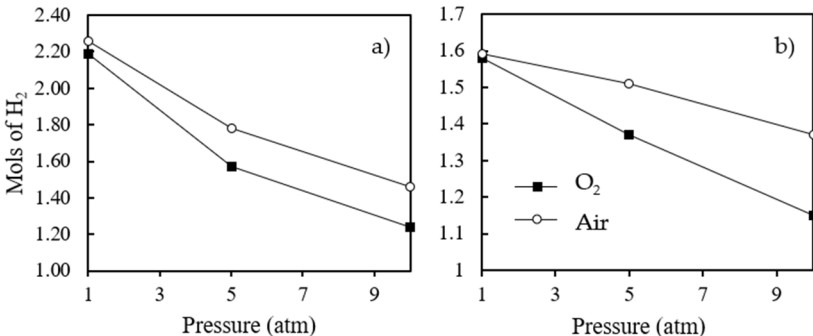

**Figure 5.** Hydrogen production at different pressures at 973 K: (**a**) OCR = 0.5; (**b**) OCR = 1.

The increase obtained using air becomes more independent of pressure as the temperature increases and, in general, when the ratio between oxygen and methane is high, a behavior also observed by Souza et al. [26], indicating that at high inlet temperatures and high ratios of oxygen to methane, hydrogen production does not depend on pressure. The average reductions in percent rise using air when raising the temperature under pressures of 1.5 and 10 atm (over the entire $O_2/CH_4$ range) were 5.3%, 13.8% and 16.5%, respectively. In the same order, these values with the increase of oxygen were 3.6%, 6.4% and 9.1%.

The greatest amount of $H_2$ formed was obtained at 1073 K, $O_2/CH_4$ = 0.5 and 1 atm, the condition shown in Figure 6. However, relevant proportions of this compound were obtained at higher pressures, such as at 5 and 10 atm, demonstrating the capacity of the process of operating in a wide range of pressures, further aided by the possibility of mitigating the negative effect of pressure on the generation of hydrogen using air.

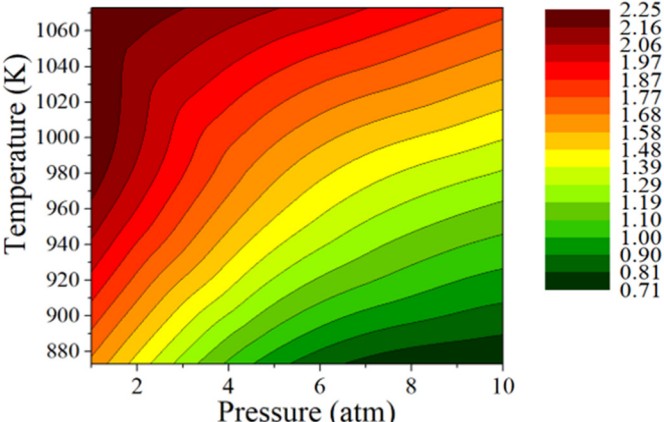

**Figure 6.** Surface graph and contour curve for $O_2/CH_4$ = 0.5 and 1 atm, using air as oxidizing agent.

Thus, the reaction using air would be advantageous in terms of moles of $H_2$ produced over the use of pure oxygen at low temperatures and moderate or high pressures for isothermal systems. Although for the reaction, the best conditions are found at high temperatures, low pressures and low $O_2/CH_4$ ratios, a region in which there is not much difference between the oxidizer used.

The results obtained at 1 atm showed that the increase in temperature is positive for the formation of $H_2$ in ratios of $O_2/CH_4$ around 0.5 and starts to be negative for proportions above that, since the synthesis gas has a negligible increase in ratios greater than 1. For $O_2/CH_4$ equal to 2.0, the amount of $H_2$ produced is negligible due to the large excess of oxygen. The noted increase of $H_2$ with temperature is due to increased methane conversion and hydrogen selectivity accompanied by thermal increases in the system. On average, for

$O_2/CH_4 = 0.5$, there were increases of 71.9% and 56.0% for the compounds of interest using air and $O_2$, respectively, when raising the temperature at different pressures. However, the rate of conversion increase starts to drop at temperatures higher than 773 K, as also reported by Yan et al. [12] and Ayabe et al. [20]. This behavior is also indicated in the approximation of the number of moles of hydrogen and synthesis gas produced for the two oxygen sources, as the temperature increases the use of air or pure $O_2$ reach similar conversions.

In Figure 7 we can see how the three parameters varying together affect the reaction in relation to the production of $H_2$ and syngas, in this case using atmospheric air as oxidizing agent.

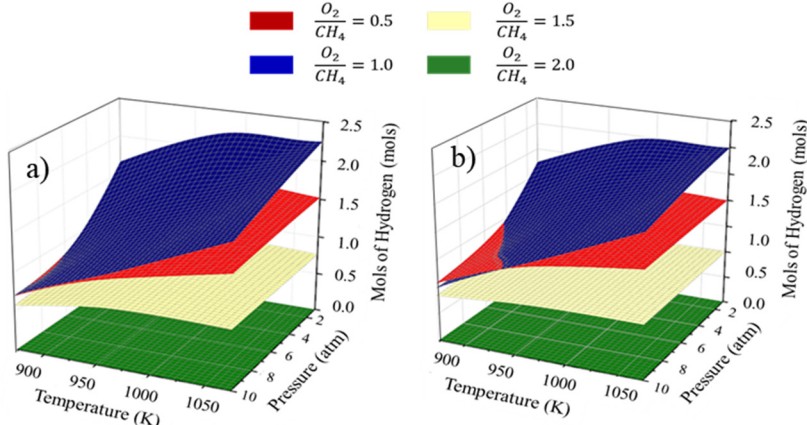

**Figure 7.** Hydrogen (**a**) and syngas (**b**) production for different reaction conditions using air as oxidizing agent.

Although the use of air increased the number of moles of $H_2$, its use decreased the fraction of hydrogen in the reformed gas at the exit, a dilution imposed by the presence of $N_2$ in the process, as observed in Figure 8. This makes it difficult to separate this compound from the gas mixture, in addition to the fact that the reactor used must have its volume expanded due to the greater amount of matter entering [26].

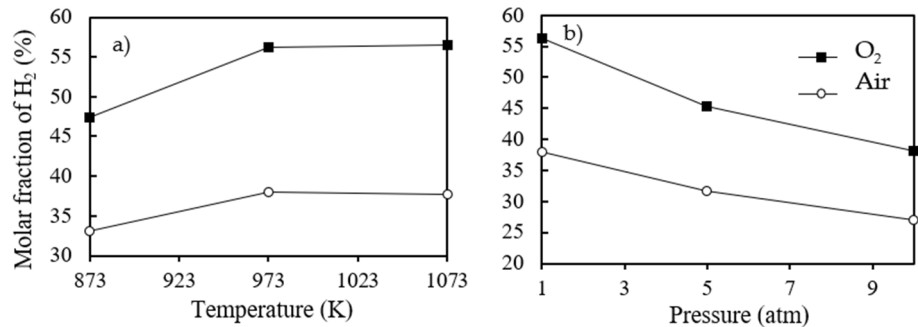

**Figure 8.** Molar fraction of hydrogen at different temperatures (**a**) and pressures (**b**) for the ratio OCR = 0.5.

On the other hand, the use of air would be interesting from the point of view of low cost regarding its availability, reduction of the temperature inside the reactor, avoiding hot spots [26] and by the production of ammonia. An additional source of nitrogen would not be necessary for the Haber–Bosch process to be able to reach ratios closer to 3:1 of $H_2$ to $N_2$, which would be ideal for the formation of $NH_3$, subject to changes in the proportion of oxygen in the air [8].

### 3.3. Thermodynamic Analysis for Adiabatic System Operation

The Gibbs energy minimization method proved to be reasonable to predict the composition of the process outlet gas, considering isothermal operating conditions. However,

most reactors for the autothermal reforming process work under adiabatic conditions, not under constant temperature [3]. To evaluate the final temperature of the system, simulations were carried out based on entropy maximization, considering adiabatic conditions for the process. The ranges of the parameters used did not change in relation to the study of the system under isothermal conditions (i.e., the minimum energy Gibbs model previously presented).

In order to validate the thermodynamic entropy maximization model, the results obtained by the model were compared with experimental data obtained in the literature. Table 2 shows the comparison between Sayar's experimental data [3] and the data obtained in this work using the entropy maximization model. The mean deviation was around 2.9%, demonstrating good ability of the model used to predict the system equilibrium temperature with good reliability.

**Table 2.** Comparison between Sayar's experimental data [3] and those of the present work using entropy maximization model.

| | **Parameters** | | | |
|---|---|---|---|---|
| Condition 1 OCR = 0.4; P = 1 atm; SCR = 3.0 | $T_i$ (K) | $T_f^{exp}$ (K) | $T_f^{calc}$ (K) | Deviation (%) |
| | 723 | 906 | 894 | 1.3 |
| Condition 2 OCR = 0.5; P = 1 atm; SCR = 3.83 | $T_i$ (K) | $T_f^{exp}$ (K) | $T_f^{calc}$ (K) | Deviation (%) |
| | 723 | 1031 | 945 | 8.3 |

Where $T_i$ inlet temperature, $T_f^{exp}$ is the experimental final temperature determined by Sayar [3] and $T_f^{calc}$ is the final temperature calculated using entropy maximization model.

Figure 9 shows the final system temperature for different inlet temperatures, pressures and $O_2/CH_4$ ratios in the system feed stream. The reduction of the final temperature when using atmospheric air in relation to pure oxygen is remarkable due to the fact that part of the energy used to heat the system is added to the inert gas flow [25], an effect also observed by Freitas and Guirardello [15]. In summary, there was on average, a 5% temperature reduction using air compared to the introduction of pure oxygen, which is favorable to avoid hot spots in the reactor. Furthermore, the increase in the outlet temperature of the gaseous stream is noticeable with the addition of oxygen in relation to methane in the system, as shown in Figure 10, which was already expected due to the concomitant predominance of exothermic reactions.

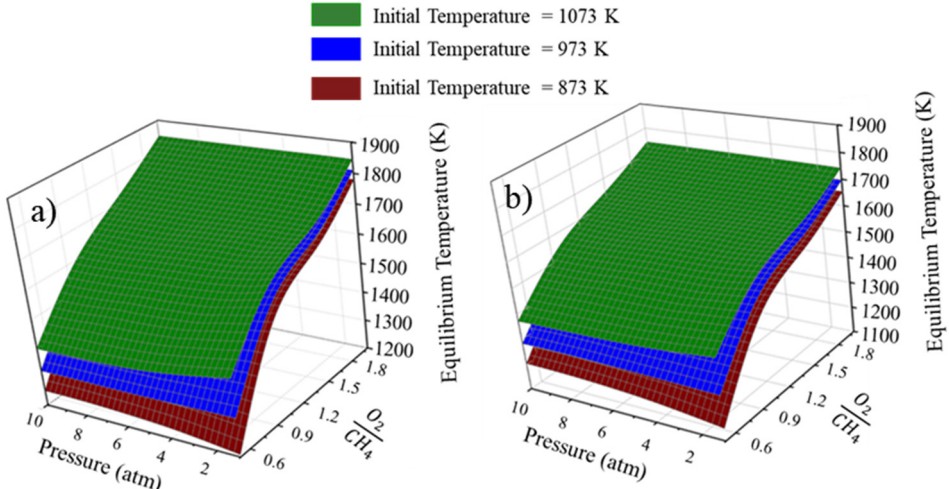

**Figure 9.** Equilibrium temperatures of autothermal reforming of methane systems using $O_2$ (**a**) and air (**b**) as oxidizing agent.

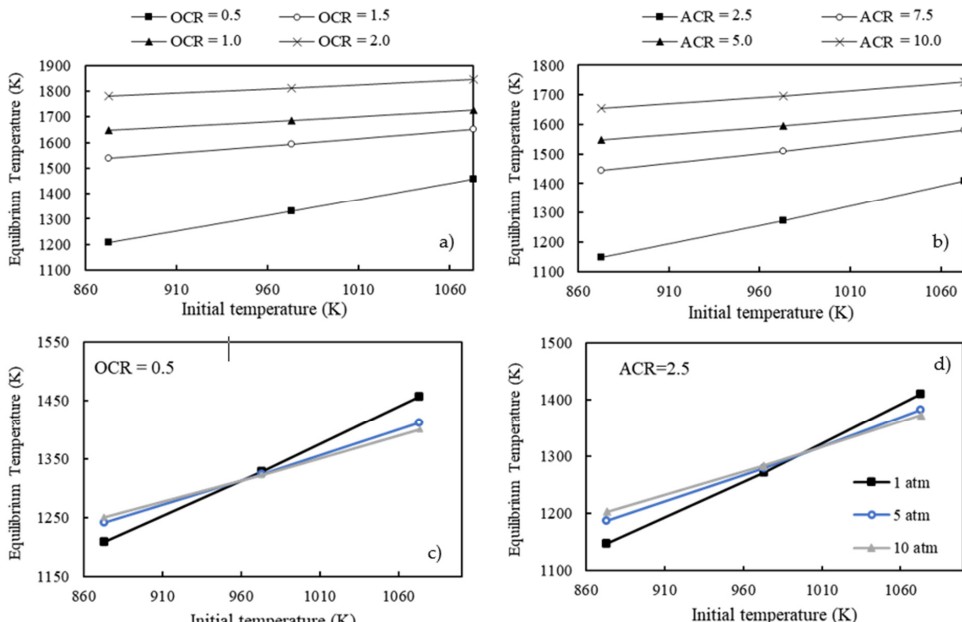

**Figure 10.** Variation of equilibrium temperature at different reaction conditions: (**a**) as a function of initial temperature for fixed values of OCR = 0.5–2; (**b**) as a function of initial temperature for fixed values of OCR = 2.5–10; (**c**) as a function of initial temperature for PCR = 0.5 for fixed pressure; (**d**) as a function of initial temperature for PCR = 2.5 for fixed pressure.

Figure 10 presents the behavior of the system in relation to different inlet pressures and temperatures, again observing the drop in temperature using air. For low initial temperatures, an increase in the final temperature is observed as the pressure increases. This trend is valid throughout the temperature range for low oxygen proportions, but reverses when its presence is intensified. When the $O_2$ ratio is half that of $CH_4$, this inversion occurs at about 973 K and holds for higher ratios.

Furthermore, it was found that as the amount of oxygen increases proportionally, the influence of pressure on the process equilibrium temperature decreases. Figure 11 illustrates the self-thermicity of the system, a behavior that was repeated for all the conditions studied, indicating that the system manages to maintain itself without the supply of additional energy.

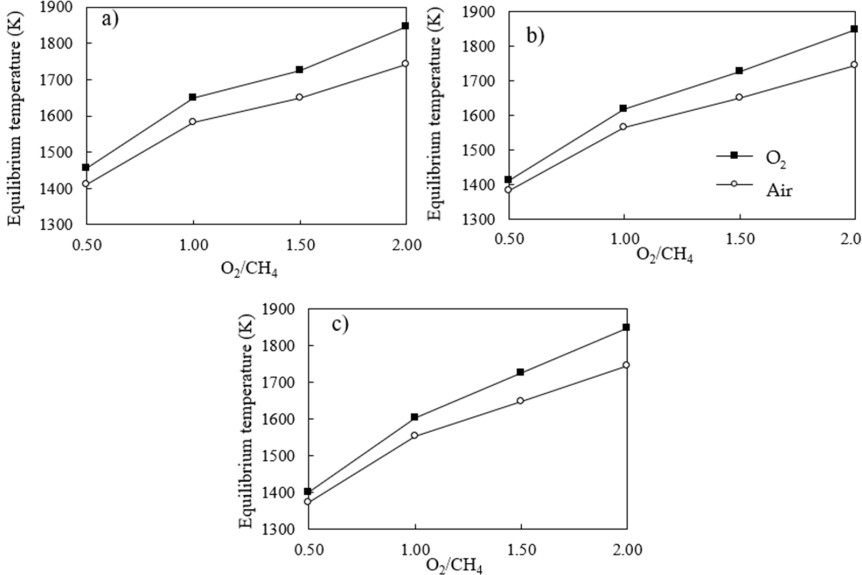

**Figure 11.** Influence of the $O_2/CH_4$ molar ratio at 1073 K and 1 atm (**a**), 5 atm (**b**), and 10 atm (**c**).

## 4. Conclusions

A complete thermodynamic analysis of hydrogen and syngas production from methane ATR was performed. The use of models based on Gibbs energy minimization and entropy maximization provided satisfactory predictions for the composition and equilibrium temperature of the ATR system. There was substantial agreement between the simulation results and the literature data for the equilibrium composition of $H_2$ and synthesis gas (deviations ranged from 1% to 5%) and in relation to the equilibrium temperature (average deviation of 2.9%), thus indicating the reliability of the models tested in relation to the experimental data of this system. These data represent an advance in the descriptive quality of this type of thermodynamic model in relation to data previously published in the literature.

The production of hydrogen and synthesis gas increased proportionally with temperature; however, a correlation of this production with the molar ratio of oxygen at the system inlet was observed. The autothermicity of the process was also verified for all reaction conditions studied. Greater increments in the final temperature were observed for the region of high oxidant concentration in the feed stream of the process.

The use of air as an oxidizer boosted the formation of the compounds of interest compared to use of pure $O_2$, especially at high operating pressures. However, high temperatures and oxygen ratios minimized this effect. The better reaction conditions involve high temperatures, low pressures and reduced $O_2/CH_4$ ratios in the feed stream. Finally, the use of air reduced the final temperature of the system by about 5% due to the nitrogen present, which could minimize the negative effects of high temperatures in a reactor conducting this reaction.

In general, the results allowed us to infer that the ATR of methane using atmospheric air as oxidizing agent presents thermodynamic advantages in relation to the use of pure oxygen, both in the production of hydrogen and synthesis gas, and in the temperature control of the process.

**Author Contributions:** A.C.D.d.F., R.G. and M.H.S.C., project proposal; A.C.D.d.F., R.G. and M.H.S.C., methodology development; M.H.S.C., research and validation; J.M.d.S.J., Í.A.M.Z., M.H.S.C., E.É.X.G.F. and A.C.D.d.F., constant evaluation of results; A.C.D.d.F., R.G. and A.D.S.V., supervision and guidance throughout the development of the article. All authors have read and agreed to the published version of the manuscript.

**Funding:** This research received no external funding.

**Institutional Review Board Statement:** Not applicable.

**Informed Consent Statement:** Not applicable.

**Data Availability Statement:** The data presented throughout the text were obtained through thermodynamic methodologies of Gibbs energy minimization and entropy maximization. The data used for validation were obtained from texts indicated throughout the text, so the presented data can be reproduced.

**Acknowledgments:** The authors would like to thank the entire faculty of the Federal University of Maranhão and State University of Campinas for their contribution to the personal and professional development of countless lives and to all the professors who support the development of society. We thank CAPES (Higher Education Personnel Improvement Coordination, Brazil) for the financial incentive to the institutions.

**Conflicts of Interest:** The authors declare no conflict of interest.

## Nomenclature

| | |
|---|---|
| $B$ | Second coefficient of the virial |
| $B_{ij}$ | Second coefficient of the virial for mixture |
| $\phi_i$ | Fugacity coefficient of component $i$ |
| $\hat{\phi}_i$ | Fugacity coefficient of component $i$ in mixture |

| | |
|---|---|
| $R$ | Universal gas constant |
| $G$ | Gibbs energy |
| $H_i^k$ | Enthalpy of component $i$ in phase $k$ |
| $H_i^0$ | Enthalpy of component $i$ in the standard state |
| $H^0$ | Total enthalpy |
| $T$ | Temperature |
| $P$ | Pressure |
| $S_i^k$ | Component $i$ entropy in phase $k$ |
| $S_i^0$ | Entropy of component $i$ in the standard state |
| $n_i^k$ | Number of moles of component $i$ in phase $k$ |
| $n_i^0$ | Number of moles in standard state |
| $a_{mi}$ | Number of atoms of element $i$ in component $m$ |
| $NC$ | Number of components |
| $NE$ | Number of elements |
| $\mu_i^k$ | Chemical potential of component $i$ in phase $k$ |
| $y_i$ | Molar fraction of gases |
| $X_{CH_4}$ | Methane conversion |
| $x_i^{lit.}$ | Literature value |
| $x_i^{calc.}$ | Calculated value |
| $S_{H_2}$ | Hydrogen selectivity |
| $n_{CH_4,in}$ | Number of moles of $CH_4$ in feed stream |
| $n_{CH_4,out}$ | Number of moles of $CH_4$ in outlet stream |
| $H_2,out$ | Number of moles of $H_2$ on outlet stream |
| $n_{CO,out}$ | Number of moles of CO on outlet stream |
| $n_{CO_2,out}$ | Number of moles of $CO_2$ on outlet stream |
| Superscripts | |
| $g$ | Gas phase |
| $l$ | Liquid phase |
| $s$ | Solid phase |

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
