# Peer review of "Autothermal Reforming of Methane: A Thermodynamic Study on the Use of Air and Pure Oxygen as Oxidizing Agents in Isothermal and Adiabatic Systems"

_methane, doi:10.3390/methane2040026_

Round 1

Reviewer 1 Report

The manuscript presents a nice work on methane conversion;  looks like the authors use a big brush, which may diluted the main points of this manuscript. There is a need to refine and focus the content for clarity and probably delete some content. The figures should be improved for better readability. If the following issues can be carefully addressed, I would recommend the publication of this work at Methane.

Specific Points:

Figure Resolution: The resolution of several figures (e.g., Figure 1, Figure 6, 7, 9, etc.) should be enhanced to improve clarity and readability.

Figure 2: For Figure 2, it's suggested to use different symbols to distinguish between Ayabe's work and the current study for better differentiation.

Figure 3: Figure 3 appears compressed and some words are obscured by the legend. This should be corrected.

Figure 4: In the caption for Figure 4, reference to (c) should be clarified or corrected.

Figure Captions: Most figure captions are concise and would benefit from more detailed information. Additionally, for figures with multiple panels, it's advisable to label them (a), (b), (c), etc., for better differentiation, as demonstrated in Figure 10.

Abstract and Conclusion: Both the abstract and conclusion sections are lengthy; it is recommended to condense them for greater conciseness.

To appeal broad attention, the most recent publication regarding the conversion of methane is recommended to be cited: https://www.sciencedirect.com/science/article/pii/S0196890422000413

Addressing these points will help enhance the manuscript's quality, clarity, and overall appeal to a broader audience.

Good

Author Response

Dear reviewer, We thank you for your collaboration with our work. Attached is the response letter to the points indicated.

Reviewer 2 Report

Paper presents interesting and useful results for researchers in this field to select the most promising conditions for hydrogen and syngas generation from methane/natural gas. So it can be accepted for publication. English is to be slightly polished at proofsreading stage.

1. What is the main question addressed by the research?  

The main question  is selection of the most promising conditions (temperature, pressure and feed composition) for the processes of autothermal  reforming of methane.

2. Do you consider the topic original or relevant in the field? Does it  address a specific gap in the field?

Original aspect of research is analysis of the reforming in adiabatic conditions which was earlier not considered

3. What does it add to the subject area compared with other published  material? 

Paper added new results related to analysis of processes in adiabatic conditions.  Especially  useful is conclusion that autothermal reforming  of methane using atmospheric air as oxidizing agent presents thermodynamic advantages in relation  to the use of pure oxygen, both in the production of hydrogen and synthesis gas, and in the temperature control of the process.

4. What specific improvements should the authors consider regarding the   methodology? What further controls should be considered?   

In future authors are advised to analyze  autothermal reforming of methane in reactors equipped with monolithic catalysts presented in a lot of papers, where it was shown that oxygen is mainly consumed in the inlet part of the catalyst mainly for methane combustion. It results in a sharp temperature gradient along the monolith length depending upon the type of supported active component along with gradients of products concentrations (see, i.e. M. Hettel et al. / Catalysis Today 216 (2013) 2– 10). In main part of the monolith steam and   dry methane reforming occurs. 

5. Are the conclusions consistent with the evidence and arguments presented  and do they address the main question posed?

Yes, conclusions are proper and sound.

6. Are the references appropriate?

Yes, they are appropriate

7. Please include any additional comments on the tables and figures.

Quality of some figures is to be improved by providing a better resolution.

English is to be slightly polished at proofsreading stage

Author Response

(The authors gave the same response as above.)

Reviewer 3 Report

This work presents a thermodynamic computation on autothermal reforming. The authors have to improve the writing of the manuscript and provide more details about the novelty compared to their previous work and literature before further consideration. 

1) I recommend that the first part of the abstract be rewritten for better explanation avoiding parentheses.

2) The authors did not mention the steam-to-methane ratio in the abstract although they mentioned all other variables. Have the authors analyzed the effect of varying this ratio? Also, I suggest that "mol/mol" should be omitted. 

3) In the abstract, the ratio "oxygen to methane" was given, but " air to methane" was not given.

4) The abstract should be rewritten considering only the main conducted work and results. For instance, I don't think using the word "ideal" is appropriate, also, use "hydrogen production" or "synthesis gas" production. the term "possible negative effects" should be explained.

5) The authors should explain why they used entropy maximization and Gibbs energy minimization at specific operating conditions.

6) This statement is not accurate "high content of hydrogen and low carbon monoxide content (syngas)." The definition of Syngas should be revised.

7) Page 2, line 51, "autothermal reformer" rather than "autothermal reform."

8) Use ATR instead of AMR for better consistency. 

9) On page 2, line 45, the author mentions that 47% of hydrogen is produced from natural gas, but they mention that 75% is from SMR. This is not consistent. Some papers indicate better consistency mentioning that around half of hydrogen produced from SMR: doi.org/10.1016/j.ijhydene.2021.04.032

10) It seems that there is a high similarity between this work and other previous works conducted by the authors. What is the novelty of this work compared to other previous studies? 

11) In line 101, I suggest using "as oxidizer" instead of "as the oxygen source."

12) In the last paragraph of the introduction, the authors should highlight the novelty of the manuscript compared to other papers in the literature. They also have to avoid redundancy in this part of the introduction section.

13) The literature review part is not sufficient and needs to be improved, The following references on ATR analysis are suggested to the authors: doi.org/10.1016/j.applthermaleng.2022.119140 , doi.org/10.1016/j.ces.2023.118987 , doi.org/10.1016/j.ijhydene.2022.12.217 , 

14) For a better explanation of the model, I suggest the authors provide a nomenclature table where they illustrate each symbol's description and unit. 

15) For the "Entropy maximization: adiabatic reactors", I suggest the authors provide a short introduction describing the calculation procedure. Also, why did the authors consider it to be an adiabatic reactor at this section? It is also should be considered as adiabatic for the Gibbs energy minimization method. 

16) The authors used GAMS and CONOPT3 Solver. What was the purpose of each tool in the thermodynamic analysis?

17) For Figure 1, I suggest providing the data on the bar chart.

18) Provide more explanation for Fig. 2. Also, I do not recommend using circles as simbols as they don't reveal the contrast between experimental and numerical results. 

19) Most figures are not well presented. The description of each graph in the figure should be added. For example, in Figure 4, figure C was not provided in the figure title. 

For example: Autothermal reform.

Author Response

(The authors gave the same response as above.)

Round 2

Reviewer 1 Report

Although the authors tried to highlight what they have changed in the new version of the manuscript, there are several concerns that need to be addressed, please provide a point-to-point response to reviewer comments. For example, I suggested that authors use different symbols to differentiate Ayabe's work and this work in Figure 2, but no actions were taken. I cannot list all of them out here again.

Also, please specify what kind of changes you have made, which is helpful for reviewers to track your revision.

Good

Author Response

Attached, we present the letter with the comments addressed.

Thank you for your attention!

Reviewer 3 Report

The manuscript was improved considering the provided comments, it can be considered for publication

Author Response

Thank you very much for all the comments and for approving our text!
